# Exploring the Potent Roles of an Internally Translated Truncated Connexin-43 Isoform

**DOI:** 10.3390/biology13121046

**Published:** 2024-12-13

**Authors:** Mario Maalouf, Adelaide T. Gaffney, Bridger R. Bell, Robin M. Shaw

**Affiliations:** Nora Eccles Harrison Cardiovascular Research and Training Institute, University of Utah, Salt Lake City, UT 84132, USA

**Keywords:** Connexin43, GJA1-20k, trafficking, cytoskeleton, mitochondria

## Abstract

Connexins are membrane proteins forming gap junctions essential for intercellular communication. Connexin 43 (Cx43), encoded by *GJA1*, is the most abundant connexin, especially in the heart. Cx43-based gap junctions enable the direct intercellular transport of ions, critical for cardiac rhythm by synchronizing electrical impulses between heart cells. The isoform GJA1-20k, derived from an internal translation start site of the same *GJA1* mRNA, has important roles beyond cell-to-cell communication. GJA1-20k aids in Cx43 trafficking to specific membrane subdomains and is crucial for cytoskeletal organization and mitochondrial stability. GJA1-20k also influences mitochondrial distribution and promotes mitochondrial fission, helping cells manage oxidative stress. Reduced GJA1-20k levels are associated with disrupted Cx43 trafficking and mitochondrial dysfunction, contributing to cardiovascular diseases such as arrhythmias and heart failure. GJA1-20k’s potent roles in regulating the cytoskeleton, Cx43 transport, and mitochondrial homeostasis makes it a promising therapeutic target.

## 1. Introduction

Connexins are a family of transmembrane proteins that form membrane channels. In mammals, there are up to 21 unique members of different gene origin making up this family. All connexins share a similar structural organization, including four transmembrane domains, as well as two extracellular and three cytosolic subdomains with the N- and C-terminal ends facing the cytosol [1,2,3]. While the N-terminus is mostly preserved between different connexins, the C-terminal cytosolic tail can vary both in length and composition [1,4]. Six connexin proteins oligomerize to form hexameric hemichannels known as connexons. Intercellular channels, known as gap junctions, are formed by connexon–connexon docking, with one connexon from each adjacent cell forming a serial pair and complete channel. Gap junctions serve essential roles in intercellular communication [5]. Once formed, gap junctions allow for direct exchange of small molecules between the cytosol of communicating cells. Gap junctions are identified by the origin of the connexin proteins which form the channel. Connexin 43 (Cx43) proteins, encoded by the gene *GJA1*, are one of the most abundantly expressed and heavily studied members of the connexin family [5].

## 2. Cx43 Physiology

Cx43-based connexons assist in rapid low-resistance intercellular communication by promoting direct cytosol-to-cytosol exchange [6]. At the membrane, connexon hemichannels of one cell dock with an adjacent cell’s hemichannel, forming a gap junction channel [7]. These channels allow hydrophilic substances, molecules, and ions smaller than 1 kDa to freely and directly pass between the cytosol of cells [8]. In healthy hearts, these exchanges allow for the rapid propagation of action potential transmission between adjacent cardiomyocytes, synchronizing each and every heartbeat into an effective pumping activity [7].

Beyond their primary role in gap junction formation, lone connexons are able to participate in autocrine and paracrine signaling [7]. Immune cells travel in circulation and migrate in tissue, rarely forming stable contact with other cells. The hemichannels Cx43 forms allow for communication and exchange with nearby, non-adjacent cells. Cx43 connexons on lymphocytes release ATP, glutamate, prostaglandins, NAD^+^, and glutathione for interaction with nearby cells [9,10,11,12,13]. Additionally, Cx43 has also been found in extracellular vesicles (EVs) released from epithelial cells, endothelial cells, cardiomyocytes, organotypic heart cultures, and circulating in human plasma [14,15,16]. At the surface of these EVs, Cx43 hemichannels allow for the rapid release of intraluminal contents directly into the cytoplasm of target cells. This occurs through docking with unopposed hemichannels on target cells, forming a structure similar to a gap junction, and followed by the release of their intraluminal contents [14,17]. These aforementioned communication pathways are believed to be essential for healthy development and tissue homeostasis and have been shown to play a major role in diverse pathological processes [7].

Extensive experimentation has also revealed multiple intracellular roles for Cx43 beyond its primary function in direct intercellular communication. Studies displayed the presence of Cx43 in intracellular organelles, including the nucleus and mitochondria [18,19]. In the studies, Cx43 has been found to have roles in gene transcription, development, mitochondrial homeostasis, autophagy regulation, intracellular trafficking, and long-distance (EV-mediated) intercellular communication [18,19].

The identification of several DNA- and RNA-binding motifs on the Cx43 sequence strongly suggests a direct role of Cx43 in gene expression within cells [18,20,21]. Cx43 profoundly affects the cellular transcriptome, altering the expression of genes involved in diverse biological processes, including cell adhesion and migration, cytoskeleton dynamics, cell signaling, proliferation, and differentiation [7]. At the mitochondrial membrane, full-length Cx43 has been detected in multiple cell types, including astrocytes, endothelial cells, and cardiomyocytes. These data indicate that the role of Cx43 involves the modulation of ROS generation, ATP levels, potassium influx to the mitochondrial matrix, and respiration—important roles within the context of ischemia/reperfusion injury [19,22]. Interestingly, Cx43 has a short half-life of less than two hours before it is signaled for degradation via ubiquitylation. After ubiquitylation, the protein is then internalized and catabolized [23,24].

## 3. Cx43 Dysfunction and Pathogenesis

As the most commonly expressed gap junction protein, Cx43 is found in multiple cell types across almost all tissues and organs [7,25]. As such, Cx43 has been the subject of extensive research, including genetic manipulation. Since global Cx43 knockout is lethal in early postnatal life, cell-specific Cx43 knockout mouse models have been developed to study the physiological effects of Cx43 in vivo [26]. For instance, endothelial cell-specific Cx43 knockout results in hypotension and bradycardia in mice [27]. In smooth-muscle cells, Cx43 knockout leads to defective remodeling in response to vascular injury [28]. Mice have impaired glucose tolerance following exposure to a high-fat diet upon hepatocyte-specific Cx43 knockout [29]. In cardiac macrophages, Cx43 deletion resulted in delayed atrioventricular conduction [30]. In osteoblasts and osteoclasts, bone mineralization and homeostasis are affected [31]. Slowed conduction velocity and sudden arrhythmic death occurred in cardiomyocytes [32]. Moreover, the unregulated opening of Cx43 hemichannels has also been associated with several dermatological and cardiovascular diseases [33,34].

## 4. Role of Cx43 in the Heart

Cx43 is the most abundant connexin protein found in the myocardium and the dominant ventricular coupling protein [35,36]. In ventricular cardiomyocytes, following translation, Cx43 is targeted to the intercalated disks, where apposing hemichannels between two adjacent cardiomyocytes form gap junctions to facilitate action potential propagation [25,35]. The localization of Cx43 at intercalated disks promotes anisotropic propagation of the electrical impulse throughout the myocardium [37].

Mutations and deficiencies in either Cx43 expression or gap junction formation are associated with various cardiovascular pathologies [38,39,40,41,42,43,44]. Ischemia/reperfusion injury in mouse models causes a remodeling and pathological opening of Cx43 hemichannels in non-junctional regions of cardiomyocytes. Mislocalization of Cx43 hemichannels can immediately lead to the development of life-threatening arrhythmias due to disrupted electrical communication and myocyte function [35]. This is corroborated through images of diseased hearts stained for Cx43, which display an altered Cx43 distribution pattern in the ventricular myocardium [36]. For instance, following a myocardial infarction, Cx43 is localized to regions near the lateral membrane of cardiomyocytes rather than the intercalated disks [36]. Other studies have shown that reducing Cx43 expression results in uncoupling between transmural cardiac muscle layers. Uncoupling leads to slowed cardiac conduction and dispersion of repolarization between the epicardial and myocardial layers [6]. Alterations to the protein, changes in the distribution of the protein, or fluctuations in the amount of Cx43 can also cause cardiac electrical abnormalities and arrhythmias [36].

## 5. Internal Translation of Cx43

In addition to full-length Cx43, there are endogenously produced, internally translated C-terminal isoforms of Cx43 that are derived from the same *GJA1* mRNA through alternative translation [45,46,47,48]. Alternative translation occurs when ribosomal translation initiates at a downstream triplet rather than the first coding triplet [1]. The protein isoforms are truncated, forming smaller proteins of different lengths [7]. The *GJA1* mRNA contains six additional start codons (AUG) encoding for Methionine, downstream of the initial coding AUG. Translation can begin at any of these sites resulting in proteins that lack the Cx43 N-terminus. This allows for the translation of seven unique proteins, including the full-length Cx43. The proteins produced by this process are approximately 43, 32, 29, 26, 20, 11, and 7 kDa in size [1,47]. These isoforms are expected to have different functions than their full-length counterpart due to their differing amounts of membrane domains and varying sizes [49]. Several isoforms have already produced a new understanding of gap junction trafficking, function, and regulation. Mechanisms previously understood to be carried out by full-length Cx43 have been recently attributed to these truncated isoforms [25]. As matter of fact, the cardioprotective effect Cx43 confers in ischemia/reperfusion injuries has now been attributed to GJA1-20k. Previous studies have shown that the involvement of Cx43 in ischemic preconditioning is independent of gap junction formation [50,51,52]. However, recent studies performed on mice with a reduced ability to express GJA1-20k but no effect on the expression of full-length Cx43 have challenged this concept. Despite expressing full-length Cx43, these mice suffered from severe conduction abnormalities and had increased infarct size following ischemia/reperfusion injury [24,53]. These findings emphasize the importance of GJA1-20k in potentiating the cardioprotective effect originally attributed to Cx43. The discovery of truncated isoforms helps clarify and delineate the non-canonical functions of ion channels, as well as highlighting the importance of internal translation in the emergence of these non-canonical functions.

## 6. Role of GJA1-20k

Although primarily a gap junction protein, Cx43 additionally exhibits gap junction-independent functions, ranging from cell growth, differentiation, and metabolism to migration. This has been attributed to its long cytosolic C-terminal tail, which undergoes extensive post-translational modifications [54,55]. However, the notion that full-length Cx43 is directly responsible for its gap junction-independent functions has been challenged by the discovery of the N-terminus truncated isoforms of Cx43, which share its C-terminal domain [47,48]. Through cap-independent translation, Cx43 mRNA leads to the generation of several smaller isoforms originating at internal translation initiation sites, with the most abundant isoform being GJA1-20k, whose name corresponds to its approximate molecular weight [46,47]. Despite its relatively recent discovery, GJA1-20k has been implicated in several pathways and processes, including Cx43 trafficking, cytoskeletal remodeling, and mitochondrial homeostasis [56,57,58,59].

## 7. Role of GJA1-20k in Cx43 Trafficking

Cx43 trafficking to the intercalated disks of cardiomyocytes follows the targeted delivery paradigm, whereby protein delivery is mediated by a complex interplay between the cytoskeleton and anchoring proteins, leading to targeted localization at distinct membrane subdomains [60]. Following transcription, translation, and post-translational modification, Cx43 monomers are transported to the Trans-Golgi Network (TGN), where they oligomerize into hexamers and form gap junction hemichannels [59]. From the TGN, Cx43 hemichannels are then trafficked along microtubules to reach their target destination at the membrane—specifically the intercalated disk in cardiomyocytes—through recognition of the anchoring protein complex. GJA1-20k underlies this process through remodeling actin filaments, which, in turn, determines the direction of microtubule trajectories and orients them to the corresponding target sites at the membrane [56].

The trafficking role of GJA1-20k was further corroborated through in vivo studies using the GJA1^M213L/M213L^ mouse line generated by the CRISPR/Cas9 gene editing system [24,61]. This mouse line contains a point mutation in the *GJA1* gene, leading to the substitution of methionine by leucine at codon 213 (M213L). This mutation does not affect the transcription of the *GJA1* mRNA; however, it reduces the translation of the internally translated GJA1-20k isoform without affecting the translation of full-length Cx43. Mice homozygous for the M213L mutation had significantly reduced gap junction formation and Cx43 retention in the cytosol, leading to its rapid degradation. Consequently, these mice suffered from severe conduction abnormalities despite preserved cardiac function and ultimately died within 4 weeks of age [24]. This deadly phenotype resulting from the absence of GJA1-20k further solidifies the importance of GJA1-20k for the maintenance of Cx43 trafficking and proper electrical activity in mammalian hearts.

## 8. Role of GJA1-20k in Cytoskeletal Remodeling

The current understanding is that GJA1-20k facilitates Cx43 trafficking to the membrane through modulation of the actin cytoskeleton, which in turn alters the orientation and trajectory of microtubule filaments transporting Cx43 monomers [56]. Actin remodeling is mediated through a direct interaction between GJA1-20k and actin, resulting in actin filament stabilization [56,58]. Binding occurs via the RPEL-like actin binding motif present at the C-terminus of GJA1-20k, leading to the formation of thickened actin filaments and actin puncta [58]. Although the overall effect of GJA1-20k on actin filaments results in net growth, GJA1-20k does not induce actin polymerization. Rather, GJA1-20k prevents actin depolymerization, as shown through previous studies wherein polymerized actin filaments treated with GJA1-20k are resistant to Latrunculin A (LatA), a known disruptor of actin fibers [53,56].

In addition to its indirect effect on microtubule remodeling achieved by patterning actin cytoskeletal remodeling, GJA1-20k is able to interact directly with microtubule filaments through the microtubule-binding domain found at its N-terminus. This interaction is critical for microtubule-dependent mitochondrial transport within cells [57]. Mitochondrial distribution and mobility are key factors that affect the dynamic mitochondrial network globally and individual mitochondrial homeostasis and function locally [62,63]. To that effect, GJA1-20k maintains the integrity of the mitochondrial network through microtubule-dependent trafficking of mitochondria to the cell periphery. Furthermore, through mediation of this microtubule–mitochondrion interaction, GJA1-20k prevents mitochondrial fragmentation induced by oxidative stress and preserves mitochondrial function [57].

Through its ability to remodel and organize both actin and microtubule filaments, GJA1-20k participates in the trafficking of both proteins and organelles, as exemplified by its effect on both Cx43 and mitochondria, to maintain proper intra- and intercellular function and homeostasis.

## 9. Role of GJA1-20k in Mitochondrial Homeostasis

Through its microtubule-binding domain, GJA1-20k is able to influence mitochondrial distribution [57]. However, the effect of GJA1-20k on mitochondria is not simply limited to the intracellular distribution of mitochondria but additionally includes a functional role of great importance.

Mitochondria are dynamic organelles that form a highly interconnected network which constantly undergoes fission and fusion, processes crucial for proper function, distribution, and quality control [64]. Mitochondrial fusion helps mitigate stress by mixing the contents of partially damaged mitochondria, while fission enables the removal of damaged mitochondria, serving as a checkpoint for quality control during mitochondrial division. The balance between these two processes is crucial for energy production, cellular metabolism, and the response to stress. Imbalance between these two processes is associated with various diseases, including cardiovascular ones [64,65].

GJA1-20k plays a critical role in maintaining mitochondrial homeostasis during stress through the promotion of protective mitochondrial fission. This allows for the removal of damaged mitochondria through mitophagy and for the generation of new, healthy mitochondria [49,53]. The ability of GJA1-20k to promote mitochondrial fission appears to be independent of the canonical mitochondrial fission machinery as it does not require dynamin-related protein 1 (DRP1) to induce fission [53]. Previous experiments have shown that mitochondrial fission still occurred with GJA1-20k administration despite siRNA mediated knockdown of DRP1. Moreover, GJA1-20k does not appear to affect DRP1 activity, as it had no effect on DRP1 phosphorylation, which is required for DRP1 potentiation. Taken together, these results indicate that GJA1-20k mediates mitochondrial fission in a manner independent of DRP1. Interactions with the mitochondrial fusion machinery proteins Mitofusin 1 and 2 (MFN1/2) were also assessed but similarly revealed no apparent interaction between them and GJA1-20k [53]. GJA1-20k is able to mediate mitochondrial fission through its ability to modulate the actin cytoskeleton; it recruits actin around mitochondria independently of DRP1 and induces focal constriction sites, leading to rapid fission events [53].

GJA1-20k’s ability to induce mitochondrial fission is crucial for protecting cells under stress by maintaining mitochondrial dynamics. During stress, an imbalance in these dynamics can lead to excessive reactive oxygen species (ROS) production and cellular damage. GJA1-20k helps counteract this by promoting mitochondrial fission, which reduces ROS levels and prevents damage. In vivo assays using a mouse model of ischemia/reperfusion (I/R) injury with GJA1-20k overexpression showed a significant reduction in cardiac damage and myocardial infarct size compared to controls. The study also found increased mitochondrial biogenesis and reduced ROS production, suggesting that GJA1-20k enhances the heart’s resistance to stress by preserving mitochondrial function and reducing oxidative damage [56]. By modulating these processes, GJA1-20k helps to maintain the bioenergetic function of mitochondria, reduce oxidative stress, and protect cells from apoptosis.

## 10. The Structure of GJA1-20k

A key aspect of GJA1-20k’s functionality is its diverse interactome, which includes multiple cytoskeletal elements, such as actin and tubulin, which are essential for the maintenance of cellular architecture and for the intracellular trafficking of proteins. These interactions are also crucial for the regulation of mitochondrial fission and maintenance of mitochondrial health. The identification of these binding partners has provided significant insights into the multifunctional nature of GJA1-20k, demonstrating its involvement in processes ranging from gap junction assembly to mitochondrial regulation.

The unique structure of GJA1-20k is closely tied to its functional role. The absence of the hydrophobic transmembrane domains, which are responsible for the formation of the gap junction pore upon the oligomerization of the Cx43 monomer, allows GJA1-20k to engage in alternative cellular pathways without the constraints imposed by the N-terminal domain (Figure 1A). This structural modification enables GJA1-20k to act as a chaperone for Cx43, ensuring its proper trafficking to the plasma membrane and protecting it from proteasomal degradation [24,56]. Furthermore, the truncated structure allows GJA1-20k to interact more freely with cytoskeletal proteins and mitochondrial regulators, thereby exerting its effects on cytoskeleton remodeling and mitochondrial homeostasis [49,53,57].

The ability of GJA1-20k, a truncated isoform of the Cx43 C-terminus, to act independently as a regulatory domain for several cellular processes prompts the need for understanding its biochemical and biophysical properties. Interestingly, NMR diffusion experiments performed at varying pH levels have shown that the Cx43 C-terminus has a tendency to form oligomers. These results are further corroborated through chemical cross-linking and analytical ultracentrifugation, which revealed that the Cx43 C-terminus exists primarily in a dimeric state, especially in more acidic environments (Figure 1B) [66]. Dimerization sites were additionally discovered (M281–N295, R299–Q304, S314–I327, and Q342–A348), with potential implications with respect to full-length Cx43 regulation, as these regions contained known phosphorylation sites and binding domains (Figure 1C) [66]. Although the physiological implications of GJA1-20k dimer formation have yet to be addressed, dimerization could potentially explain the functional versatility of this isoform and could further enhance its activity in specific cellular contexts, such as during the modulation of mitochondrial dynamics or cytoskeletal remodeling.

Instead of a singular GJA1-20k protein linking different binding partners together, dimerization allows for distinct populations of GJA1-20k proteins to act as an intermediary between proteins without the risk of stereotactic interference impeding this interaction. This could explain how GJA1-20k is able to simultaneously rearrange the actin cytoskeleton, pattern microtubules in parallel arrays to actin, and ensure Cx43 vesicles are transported along those microtubule highways to their corresponding membrane subdomains (Figure 2). GJA1-20k dimerization further facilitates actin recruitment around the mitochondria and the formation of constriction sites by having different GJA1-20k populations responsible for distinct roles.

The structure–function relationship of GJA1-20k is a prime example of how the structural features of a protein can dictate its interactions and functional outcomes. The ability of GJA1-20k to form these dimers would allow it to participate in a wide range of cellular processes, reflecting its adaptability and importance in maintaining cellular function under conditions of stress.

## 11. Clinical Considerations

Currently, GJA1-20k is primarily considered a trafficking protein essential for the targeted delivery of Cx43. However, emerging evidence increasingly highlights its independent roles beyond its association with its parent protein. Understanding the mechanism of action of GJA1-20k is essential for the development of novel interventions for treating different diseases.

A recent study has explored the therapeutic potential for GJA1-20k in treating a desmoglein mutant murine model of arrhythmogenic cardiomyopathy (ACM) [67]. This study showed that GJA1-20k gene therapy can reduce arrhythmic burden in this ACM mouse model by rescuing impaired Cx43 forward trafficking, independent of systolic function. GJA1-20k restores Cx43 functionality, effectively reducing the risk of lethal arrhythmias associated with ACM. These results underscore GJA1-20k’s potential as a therapeutic target in cardiac arrhythmias and related pathologies.

Beyond the heart, emerging research suggests that GJA1-20k may play critical roles in other tissues and disease contexts. Studies have documented its role in rescuing mitochondrial networks under oxidative stress and providing neuroprotection after traumatic brain injuries (TBI). GJA1-20k protects mitochondrial networks under oxidative stress by acting as an organelle chaperone, leading to mitochondrial localization at the cell periphery [57]. This interaction was shown in astrocytes post-traumatic brain injury (TBI), wherein GJA1-20k stabilized the mitochondria, facilitated their movement, and enhanced their transfer to neurons via tunneling nanotubules [68]. These findings suggest that GJA1-20k could be a therapeutic target for neurodegenerative diseases or brain injuries.

GJA1-20k has also been shown to regulate myometrial activity during labor. GJA1-20k expression is differentially regulated by progesterone receptor isoforms PR-A and PR-B which, respectively, promote and inhibit its activity. Dysregulation of GJA1-20k by hormonal perturbations may lead to preterm labor and other obstetric complications due to its downstream effects on Cx43 trafficking, gap junction formation, and myocyte connectivity, processes essential for synchronized uterine contractions [69].

The role of GJA1-20k in cancer pathology is also gaining attention. While full-length Cx43 has been widely studied as a tumor suppressor and, paradoxically, as a pro-metastatic agent in certain contexts, GJA1-20k may hold distinct regulatory functions [70,71]. Tishchenko et al. investigated the role of GJA1-20k in breast cancer cell behavior [72]. GJA1-20k was found to regulate actin organization, guide Cx43 delivery, and promote nanotubule formation, impacting proliferation, migration, and epithelial-to-mesenchymal transformation [72]. These findings indicate that GJA1-20k has a dual role in tumor suppression and progression, opening new research avenues for targeted cancer therapies.

## 12. Conclusions

The clinical relevance of GJA1-20k is particularly evident in the context of cardiovascular diseases, where its roles in Cx43 trafficking, cytoskeleton remodeling, and mitochondrial homeostasis are critical for maintaining cardiac function and protecting the heart from injury (Figure 2) [67,73]. Reduced GJA1-20k levels can lead to disrupted trafficking of Cx43, resulting in impaired gap junction communication, which is essential for the synchronized contraction of the heart muscle. This disruption can contribute to the development of arrhythmias, as the loss of functional gap junctions leads to abnormal electrical conduction and an increased susceptibility to re-entrant circuits. Additionally, the role of GJA1-20k in maintaining mitochondrial homeostasis is particularly relevant in the heart, where mitochondrial dysfunction is a key factor in the pathogenesis of ischemic heart disease and heart failure. A reduction in GJA1-20k can lead to impaired mitochondrial fission, increased oxidative stress, and the activation of cell death pathways—all of which contribute to the progression of cardiac dysfunction.

GJA1-20k represents a promising therapeutic target for the treatment of cardiovascular diseases. Strategies that aim to enhance GJA1-20k expression or mimic its activity could potentially improve cardiac function, reduce the risk of arrhythmias, and protect the heart from ischemia/reperfusion injury. Furthermore, the therapeutic potential of GJA1-20k extends beyond the heart as its roles in mitochondrial regulation and cytoskeletal dynamics may also be relevant in other tissues, particularly those subject to high metabolic demand or either physical or oxidative stress.

## Figures and Tables

**Figure 1 biology-13-01046-f001:**
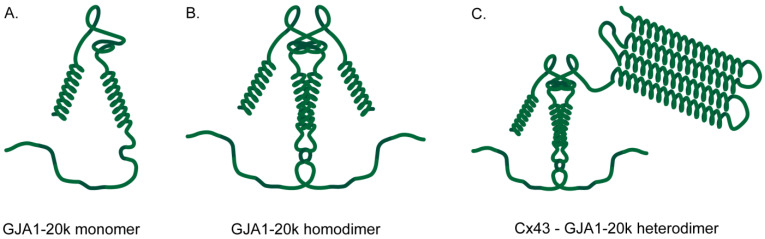
Schematic representation of (**A**) GJA1-20k monomer; (**B**) GJA1-20k homodimer; (**C**) Cx43-GJA1-20k heterodimer. Cx43: Connexin43.

**Figure 2 biology-13-01046-f002:**
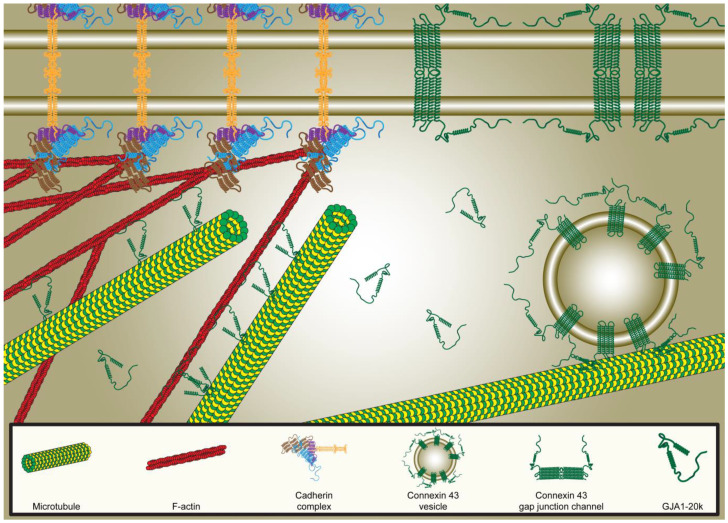
Schematic representation of targeted Cx43 delivery to the membrane mediated through interactions of GJA1-20k with the actin cytoskeleton and microtubules. Actin filaments undergo remodeling by GJA1-20k, which in turn determine the patterning and orientation of microtubules to the corresponding target sites at the membrane for proper Cx43 trafficking. Cx43: Connexin43.

## Data Availability

No new data were created or analyzed in this study. Data sharing is not applicable to this article.

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
