# Peer review of "Exploring the Potent Roles of an Internally Translated Truncated Connexin-43 Isoform"

_biology, 2024, doi:10.3390/biology13121046_

Round 1
Reviewer 1 Report
Comments and Suggestions for Authors
Comments and Suggestions for Authors
-
Abstract:
- Strengthen the focus on the novelty of GJA1-20k’s functions and its therapeutic potential.
- Include a brief mention of the clinical relevance of the findings.
-
Introduction:
- Clarify the research question or hypothesis early in the introduction.
- Provide more background on how the discovery of GJA1-20k has shifted current understanding of Cx43 functions.
-
Figures:
- Enhance the figure captions with detailed descriptions of processes and abbreviations used.
- Consider improving figure resolution for better clarity, especially for complex pathways.
-
Clinical Applications:
- Expand the discussion on how GJA1-20k could be leveraged in therapies for arrhythmias, heart failure, or ischemia/reperfusion injury.
- Address how these findings could be translated into preclinical or clinical studies.
-
Consistency and Clarity:
- Avoid repetitive explanations of GJA1-20k’s role in mitochondrial dynamics and cytoskeletal remodeling.
- Summarize these roles in a table for easier reference by readers.
-
Language:
- Simplify overly complex sentences for better readability.
- Revise minor grammatical errors and streamline sections with repetitive content.
-
Future Directions:
- Suggest potential avenues for further research, such as exploring GJA1-20k in non-cardiac tissues or as a biomarker for disease.
Author Response
We thank the reviewer for the thorough review and valuable suggestions to improve our work. We have revised the manuscript and provided answers to the reviewer’s concerns to enhance the clarity and understanding of our work. We trust that our revised manuscript has adequately addressed the reviewer’s critiques. Please find below our detailed response to each specific comment.
- Abstract: Strengthen the focus on the novelty of GJA1-20k’s functions and its therapeutic potential. Include a brief mention of the clinical relevance of the findings.
We would like to thank the reviewer for this important suggestion. We have revised the abstract accordingly to emphasize GJA1-20k’s functions, clinical relevance, and therapeutic potential.
The revised manuscript has been updated to include the following (line 20 – 35):
Abstract: Connexin 43 (Cx43) is an essential regulator in cardiovascular physiology, responsible for intercellular communication within the heart. The role of Cx43 in maintaining beat-to-beat cardiac excitation and cardiac function underscores its significance. Alterations in Cx43 expression and localization have been implicated in pathologies from sudden cardiac death to heart failure. Essential to Cx43 function is its intrinsic ability to form diverse isoforms through internal translation of GJA1 mRNA. Evidence has accumulated that GJA1-20k, the most abundant of these isoforms, is necessary for Cx43 trafficking and localization. GJA1-20k has been recognized to have a wide range of additional functions beyond directing Cx43 based intercellular communication, including cytoskeletal modulation, maintaining mitochondrial homeostasis, protecting against oxidative stress, and mediating mitochondrial preconditioning. The involvement of GJA1-20k in these processes confers it great therapeutic potential, especially in treating cardiovascular diseases such as myocardial infarction, ischemia/reperfusion injuries, and arrhythmias. Administration of GJA1-20k mitigates the underlying cellular pathophysiological disturbances that develop as a result of these diseases. Numerous studies have documented the therapeutic efficacy of GJA1-20k gene therapy in animal models of cardiovascular disease. The translational impact of these studies opens up new treatment avenues through the use of gene therapy targeting novel mechanisms of action.
- Introduction: Clarify the research question or hypothesis early in the introduction. Provide more background on how the discovery of GJA1-20k has shifted current understanding of Cx43 functions.
We thank the reviewer for their feedback. Since this is a review article, it does not center around a specific research question or hypothesis but rather aims to synthesize and discuss current knowledge on the subject. As the reviewer suggested, we have strengthened the manuscript by providing additional background on how the discovery of GJA1-20k has reshaped the understanding of Cx43 functions, offering a broader context for its significance.
The revised manuscript has been updated to include the following (line 144 – 156):
Mechanisms previously understood to be carried out by full-length Cx43 have been recently attributed to these truncated isoforms [27]. As matter of fact, the cardioprotective effect Cx43 confers in ischemia/reperfusion injuries has now been attributed to GJA1-20k. Previous studies have shown that the involvement of Cx43 in ischemic preconditioning is independent of gap junction formation [53–55]. However, recent studies performed on mice with reduced ability to express GJA1-20k but no effect on the expression of full length Cx43 have challenged this concept. Despite expressing full length Cx43, these mice suffered from severe conduction abnormalities and had increased infarct size following ischemia/reperfusion injury [26,56]. These findings emphasize the importance of GJA1-20k in potentiating the cardioprotective effect originally attributed to Cx43. The discovery of truncated isoforms helps clarify and delineate the non-canonical functions of ion channels as well as highlight the importance of internal translation in the emergence of these non-canonical functions.
- Figures: Enhance the figure captions with detailed descriptions of processes and abbreviations used. Consider improving figure resolution for better clarity, especially for complex pathways.
We thank the reviewer for their valuable feedback. We have enhanced the figure resolution to 300 dpi to ensure better clarity and have updated the captions to include detailed descriptions of the processes and explanations for all abbreviations used.
The revised manuscript has been updated to include the following (line 279 – 281, 310 – 315):

Figure 1. Schematic representation of: (A) GJA1-20k monomer; (B) GJA1-20k homodimer; (C) Cx43 – GJA1-20k heterodimer. Cx43: Connexin43

Figure 2. Schematic representation of Cx43 targeted delivery to the membrane mediated through interactions between GJA1-20k with the actin cytoskeleton and microtubules. Actin filaments undergo remodeling by GJA1-20k, which in turn determine the patterning and orientation of microtubules to the corresponding target sites at the membrane for proper Cx43 trafficking. Cx43: Connexin43
- Clinical Applications: Expand the discussion on how GJA1-20k could be leveraged in therapies for arrhythmias, heart failure, or ischemia/reperfusion injury. Address how these findings could be translated into preclinical or clinical studies.
We would like to thank the reviewer for this valuable comment. We have included in our manuscript a section detailing the clinical implications of GJA1-20k as a potential therapy. We highlighted studies that investigated the role of GJA1-20k in the context of arrhythmogenic cardiomyopathy, traumatic brain injury, and cancer. We appreciate the reviewer’s concern for further discussion of therapies and preclinical studies; however, we are not currently able to disclose further information regarding the therapeutic potential of GJA1-20k in treating cardiovascular diseases and cardiomyopathies as it is an active area of work in our lab. We are optimistic that our current work will provide a more expansive response to these questions in the future. We aim to maintain a balance between providing a comprehensive overview and respecting the publication process for ongoing research. Additionally, due to a high demand from other reviewers for more detailed descriptions of the molecular mechanisms of GJA1-20k and its effects on cellular machinery, we would like to allow the therapeutic role of GJA1-20k to be covered in a later review.
The revised manuscript has been updated to include the following (line 316 – 352):
- Clinical Considerations
Currently, GJA1-20k is primarily considered a trafficking protein essential for targeted delivery of Cx43. However, emerging evidence increasingly highlights its independent roles beyond its association with its parent protein. Understanding the mechanism of action of GJA1-20k is essential for the development of novel interventions for treating different diseases.
A recent study has explored the therapeutic potential for GJA1-20k in treating a desmoglein mutant murine model of arrhythmogenic cardiomyopathy (ACM) [74]. This study showed that GJA1-20k gene therapy can reduce arrhythmic burden in this ACM mouse model by rescuing impaired Cx43 forward trafficking, independent of systolic function. GJA1-20k restores Cx43 functionality, effectively reducing the risk of lethal arrhythmias associated with ACM. These results underscore GJA1-20k’s potential as a therapeutic target in cardiac arrhythmias and related pathologies.
Beyond the heart, emerging research suggests that GJA1-20k may play critical roles in other tissues and disease contexts. Studies have documented its role in rescuing mitochondrial networks under oxidative stress and providing neuroprotection after traumatic brain injuries (TBI). GJA1-20k protects mitochondrial networks under oxidative stress by acting as an organelle chaperone, leading to mitochondrial localization at the cell periphery [63]. This interaction was shown in astrocytes post-traumatic brain injury (TBI) wherein GJA1-20k stabilized mitochondria, facilitated their movement, and enhanced their transfer to neurons via tunneling nanotubules [75]. These findings suggest that GJA1-20k could be a therapeutic target for neurodegenerative diseases or brain injuries.
GJA1-20k has also been shown to regulate myometrial activity during labor. GJA1-20k expression is differentially regulated by progesterone receptor isoforms PR-A and PR-B which respectively promote and inhibit its activity. Dysregulation of GJA1-20k by hormonal perturbations may lead to preterm labor and other obstetric complications due to its downstream effects on Cx43 trafficking, gap junction formation, and myocyte connectivity, processes essential for synchronized uterine contractions [76].
The role of GJA1-20k in cancer pathology is also gaining attention. While full-length Cx43 has been widely studied as a tumor suppressor and, paradoxically, as a pro-metastatic agent in certain contexts, GJA1-20k may hold distinct regulatory functions [77,78]. Tishchenko et al. investigated the role of GJA1-20k in breast cancer cell behavior [79]. GJA1-20k was found to regulate actin organization, guide Cx43 delivery, and promote nanotubule formation, impacting proliferation, migration, and epithelial-to-mesenchymal transformation [79]. These findings indicate that GJA1-20k has a dual role in tumor sup-pression and progression, opening new research avenues for targeted cancer therapies.
- Consistency and Clarity: Avoid repetitive explanations of GJA1-20k’s role in mitochondrial dynamics and cytoskeletal remodeling. Summarize these roles in a table for easier reference by readers.
We greatly appreciate the reviewer’s feedback. Unfortunately, we feel that the repetition of GJA1-20k’s role in mitochondrial dynamics and cytoskeleton remodeling is unavoidable. As the principle focus of our review, the molecular mechanisms involving GJA1-20k have proven to be highly interconnected, and as such, emphasis was placed on the linking GJA1-20k’s roles in both regulating the cytoskeleton and maintaining mitochondrial homeostasis as these exact roles confer onto GJA1-20k its therapeutic potential. We greatly appreciate the reviewer for bringing this concern to our attention and have made necessary changes to highlight how its role in cytoskeletal remodeling and preserving mitochondrial homeostasis holds clinical relevance and significance.
- Language: Simplify overly complex sentences for better readability. Revise minor grammatical errors and streamline sections with repetitive content.
We thank the reviewer for their feedback. We have simplified the language used and corrected grammatical errors to improve readability and clarity.
- Future Directions: Suggest potential avenues for further research, such as exploring GJA1-20k in non-cardiac tissues or as a biomarker for disease.
We would like to thank the reviewer for this valuable comment. Our manuscript has been updated with a section covering therapeutic applications for GJA1-20k in the context of arrhythmogenic cardiomyopathy, traumatic brain injury, and cancer.
The revised manuscript has been updated to include the following (line 316 – 352):
- Clinical Considerations
Currently, GJA1-20k is primarily considered a trafficking protein essential for targeted delivery of Cx43. However, emerging evidence increasingly highlights its independent roles beyond its association with its parent protein. Understanding the mechanism of action of GJA1-20k is essential for the development of novel interventions for treating different diseases.
A recent study has explored the therapeutic potential for GJA1-20k in treating a desmoglein mutant murine model of arrhythmogenic cardiomyopathy (ACM) [74]. This study showed that GJA1-20k gene therapy can reduce arrhythmic burden in this ACM mouse model by rescuing impaired Cx43 forward trafficking, independent of systolic function. GJA1-20k restores Cx43 functionality, effectively reducing the risk of lethal arrhythmias associated with ACM. These results underscore GJA1-20k’s potential as a therapeutic target in cardiac arrhythmias and related pathologies.
Beyond the heart, emerging research suggests that GJA1-20k may play critical roles in other tissues and disease contexts. Studies have documented its role in rescuing mitochondrial networks under oxidative stress and providing neuroprotection after traumatic brain injuries (TBI). GJA1-20k protects mitochondrial networks under oxidative stress by acting as an organelle chaperone, leading to mitochondrial localization at the cell periphery [63]. This interaction was shown in astrocytes post-traumatic brain injury (TBI) wherein GJA1-20k stabilized mitochondria, facilitated their movement, and enhanced their transfer to neurons via tunneling nanotubules [75]. These findings suggest that GJA1-20k could be a therapeutic target for neurodegenerative diseases or brain injuries.
GJA1-20k has also been shown to regulate myometrial activity during labor. GJA1-20k expression is differentially regulated by progesterone receptor isoforms PR-A and PR-B which respectively promote and inhibit its activity. Dysregulation of GJA1-20k by hormonal perturbations may lead to preterm labor and other obstetric complications due to its downstream effects on Cx43 trafficking, gap junction formation, and myocyte connectivity, processes essential for synchronized uterine contractions [76].
The role of GJA1-20k in cancer pathology is also gaining attention. While full-length Cx43 has been widely studied as a tumor suppressor and, paradoxically, as a pro-metastatic agent in certain contexts, GJA1-20k may hold distinct regulatory functions [77,78]. Tishchenko et al. investigated the role of GJA1-20k in breast cancer cell behavior [79]. GJA1-20k was found to regulate actin organization, guide Cx43 delivery, and promote nanotubule formation, impacting proliferation, migration, and epithelial-to-mesenchymal transformation [79]. These findings indicate that GJA1-20k has a dual role in tumor sup-pression and progression, opening new research avenues for targeted cancer therapies.

Reviewer 2 Report
Comments and Suggestions for Authors
Authors review the role of GJA1-20k isoform, located at the C-terminal tail of Cx43, that has been shown to have multiple roles due to its ability to oligomerize and form higher order structures. This protein product is unique in that it is the only ion channel that can undergo internal translation. The review is easy to follow, especially due to the well designed figures, and the concise style of the paper.
The paper is timely and relevant, because a lot of emphasis in the literature is put on the cryo-EM structure of gap junctions, that deal with the transmembrane domains, and the disordered C-terminal tail is becoming marginal, or not paid enough attention to. Though it is known, that its length is comparable to that of the full, "visible" part of Cx43.
The paper will draw the interest of cardiovascular experts and protein structure researchers, as well.
The only drawback I found was the large amount of self-citation (although relevant). I wonder, if other laboratories have also studied the GJA1-20k isoform to such an extent.
Otherwise, I recommend the review for publication.
Author Response
We would like to thank the reviewer for the helpful review of our manuscript and for the many positive comments.
Authors review the role of GJA1-20k isoform, located at the C-terminal tail of Cx43, that has been shown to have multiple roles due to its ability to oligomerize and form higher order structures. This protein product is unique in that it is the only ion channel that can undergo internal translation. The review is easy to follow, especially due to the well-designed figures, and the concise style of the paper. The paper is timely and relevant, because a lot of emphasis in the literature is put on the cryo-EM structure of gap junctions, that deal with the transmembrane domains, and the disordered C-terminal tail is becoming marginal, or not paid enough attention to. Though it is known, that its length is comparable to that of the full, "visible" part of Cx43. The paper will draw the interest of cardiovascular experts and protein structure researchers, as well. The only drawback I found was the large amount of self-citation (although relevant). I wonder, if other laboratories have also studied the GJA1-20k isoform to such an extent. Otherwise, I recommend the review for publication.
We greatly appreciate the reviewer’s valuable feedback. We acknowledge the concern about the number of self-citations in the review. However, given the contributions our lab has made to the discovery and study of GJA1-20k and the still developing work done by others on the subject, we believe that this situation is unavoidable. We have highlighted work done by other groups on GJA1-20k, and expect this field to develop rapidly. Our review provides a comprehensive summary of present findings in this emerging and developing field. We trust that this review will stimulate further research and encourage other labs to explore and expand on this area of study.
The revised manuscript has been updated to include the following (line 329 – 352, 584 – 598):
Beyond the heart, emerging research suggests that GJA1-20k may play critical roles in other tissues and disease contexts. Studies have documented its role in rescuing mitochondrial networks under oxidative stress and providing neuroprotection after traumatic brain injuries (TBI). GJA1-20k protects mitochondrial networks under oxidative stress by acting as an organelle chaperone, leading to mitochondrial localization at the cell periphery [63]. This interaction was shown in astrocytes post-traumatic brain injury (TBI) wherein GJA1-20k stabilized mitochondria, facilitated their movement, and enhanced their transfer to neurons via tunneling nanotubules [75]. These findings suggest that GJA1-20k could be a therapeutic target for neurodegenerative diseases or brain injuries.
GJA1-20k has also been shown to regulate myometrial activity during labor. GJA1-20k expression is differentially regulated by progesterone receptor isoforms PR-A and PR-B which respectively promote and inhibit its activity. Dysregulation of GJA1-20k by hormonal perturbations may lead to preterm labor and other obstetric complications due to its downstream effects on Cx43 trafficking, gap junction formation, and myocyte connectivity, processes essential for synchronized uterine contractions [76].
The role of GJA1-20k in cancer pathology is also gaining attention. While full-length Cx43 has been widely studied as a tumor suppressor and, paradoxically, as a pro-metastatic agent in certain contexts, GJA1-20k may hold distinct regulatory functions [77,78]. Tishchenko et al. investigated the role of GJA1-20k in breast cancer cell behavior [79]. GJA1-20k was found to regulate actin organization, guide Cx43 delivery, and promote nanotubule formation, impacting proliferation, migration, and epithelial-to-mesenchymal transformation [79]. These findings indicate that GJA1-20k has a dual role in tumor sup-pression and progression, opening new research avenues for targeted cancer therapies.
References:
- Ren, D.; Zheng, P.; Zou, S.; Gong, Y.; Wang, Y.; Duan, J.; Deng, J.; Chen, H.; Feng, J.; Zhong, C.; et al. GJA1-20K Enhances Mitochondria Transfer from Astrocytes to Neurons via Cx43-TnTs After Traumatic Brain Injury. Cell Mol Neurobiol 2022, 42, 1887–1895, doi:10.1007/s10571-021-01070-x.
- Nadeem, L.; Shynlova, O.; Mesiano, S.; Lye, S. Progesterone Via Its Type-A Receptor Promotes Myometrial Gap Junction Coupling. Sci Rep 2017, 7, 13357, doi:10.1038/s41598-017-13488-9.
- Lamiche, C.; Clarhaut, J.; Strale, P.-O.; Crespin, S.; Pedretti, N.; Bernard, F.-X.; Naus, C.C.; Chen, V.C.; Foster, L.J.; Defamie, N.; et al. The Gap Junction Protein Cx43 Is Involved in the Bone-Targeted Metastatic Behaviour of Human Prostate Cancer Cells. Clin Exp Metastasis 2012, 29, 111–122, doi:10.1007/s10585-011-9434-4.
- Boucher, J.; Balandre, A.-C.; Debant, M.; Vix, J.; Harnois, T.; Bourmeyster, N.; Péraudeau, E.; Chépied, A.; Clarhaut, J.; Debiais, F.; et al. Cx43 Present at the Leading Edge Membrane Governs Promigratory Effects of Osteoblast-Conditioned Medium on Human Prostate Cancer Cells in the Context of Bone Metastasis. Cancers (Basel) 2020, 12, 3013, doi:10.3390/cancers12103013.
- Tishchenko, A.; Azorín, D.D.; Vidal-Brime, L.; Muñoz, M.J.; Arenas, P.J.; Pearce, C.; Girao, H.; Ramón y Cajal, S.; Aasen, T. Cx43 and Associated Cell Signaling Pathways Regulate Tunneling Nanotubes in Breast Cancer Cells. Cancers (Basel) 2020, 12, 2798, doi:10.3390/cancers12102798.

Reviewer 3 Report
Comments and Suggestions for Authors
This is a comprehensive review of GJA1-encoded Cx43, especially GJA1-20k, the most abundant isoform of all seven isoforms generated through internal translation of GJA1 mRNA, which provides new insights into the multifaceted role of Cx43 isoforms, suggesting novel avenues for therapies targeting cardiovascular diseases.
The manuscript is written well and is very interesting.
Author Response
This is a comprehensive review of GJA1-encoded Cx43, especially GJA1-20k, the most abundant isoform of all seven isoforms generated through internal translation of GJA1 mRNA, which provides new insights into the multifaceted role of Cx43 isoforms, suggesting novel avenues for therapies targeting cardiovascular diseases. The manuscript is written well and is very interesting.
We would like to thank the reviewer for the helpful review of our manuscript and for the positive comment.

Reviewer 4 Report
Comments and Suggestions for Authors
Thank you for your comprehensive review titled "Exploring the Potent Roles of an Internally Translated Truncated Connexin-43 Isoform." Your manuscript provides valuable insights into the multifaceted roles of GJA1-20k in cardiac physiology, particularly its involvement in Cx43 trafficking, cytoskeletal remodeling, and mitochondrial homeostasis. The focus on GJA1-20k's functions beyond gap junction communication contributes significantly to the understanding of cardiovascular diseases and potential therapeutic targets.
To enhance the clarity and impact of your review, I offer the following suggestions:
1. While you provide an overview of GJA1-20k's roles, including more detailed descriptions of the molecular mechanisms would strengthen the review. For instance, elaborating on how GJA1-20k interacts with specific cytoskeletal proteins and the downstream effects on Cx43 trafficking could provide deeper understanding.
2. Discuss any known interactions between GJA1-20k and proteins directly involved in mitochondrial fission and fusion (e.g., Mfn1/2, OPA1, Drp1), even if it's to highlight differences or the independence of GJA1-20k's pathway from these proteins.
3. Expand on the potential physiological significance of GJA1-20k dimerization. Speculate on how dimer formation might influence its interactions and functions within cells.
4. Suggest areas for future research, such as unexplored functions of GJA1-20k, its role in other cell types or tissues, and how understanding its mechanisms could lead to novel interventions for cardiovascular diseases.
5. While the English is generally clear, a thorough proofreading could address minor grammatical errors or awkward phrasings to further improve readability.
Author Response
We thank the reviewer for the thorough review and valuable suggestions to improve our work. We have revised the manuscript and provided answers to the reviewer’s concerns to enhance the clarity and understanding of our work. We trust that our revised manuscript has adequately addressed the reviewer’s critiques. Please find below our detailed response to each specific comment.
Thank you for your comprehensive review titled "Exploring the Potent Roles of an Internally Translated Truncated Connexin-43 Isoform." Your manuscript provides valuable insights into the multifaceted roles of GJA1-20k in cardiac physiology, particularly its involvement in Cx43 trafficking, cytoskeletal remodeling, and mitochondrial homeostasis. The focus on GJA1-20k's functions beyond gap junction communication contributes significantly to the understanding of cardiovascular diseases and potential therapeutic targets.
To enhance the clarity and impact of your review, I offer the following suggestions:
- While you provide an overview of GJA1-20k's roles, including more detailed descriptions of the molecular mechanisms would strengthen the review. For instance, elaborating on how GJA1-20k interacts with specific cytoskeletal proteins and the downstream effects on Cx43 trafficking could provide deeper understanding.
We thank the reviewer for this important feedback. While we appreciate the suggestion to include more detailed descriptions of the molecular mechanisms, we are unable to expand significantly on this aspect, as the current work for how GJA1-20k interacts with the cytoskeleton to promote Cx43 trafficking is not yet published. We have included work that is current in press (reference 64) while still aiming to maintain a balance between providing a comprehensive overview and respecting the publication process for ongoing research.
- Discuss any known interactions between GJA1-20k and proteins directly involved in mitochondrial fission and fusion (e.g., Mfn1/2, OPA1, Drp1), even if it's to highlight differences or the independence of GJA1-20k's pathway from these proteins.
We thank the reviewer for their insightful suggestion. We have incorporated into our manuscript a discussion on GJA1-20k and proteins involved in mitochondrial fission and fusion. While interactions with canonical fission/fusion proteins have not been conclusively demonstrated, we have highlighted the independence of GJA1-20k’s pathway, offering a clearer understanding of its unique role in mitochondrial dynamics.
The revised manuscript has been updated to include the following (line 237 – 249):
The ability of GJA1-20k to promote mitochondrial fission appears to be independent of the canonical mitochondrial fission machinery as it does not require dynamin-related protein 1 (DRP1) to induce fission [56]. Previous experiments have shown that mitochondrial fission still occurred with GJA1-20k administration despite siRNA mediated knockdown of DRP1. Moreover, GJA1-20k does not appear to affect DRP1 activity as it had no effect on DRP1 phosphorylation which is required for DRP1 potentiation. Taken together, these results indicate that GJA1-20k mediates mitochondrial fission in a manner independent of DRP1. Interactions with the mitochondrial fusion machinery proteins Mitofusin 1 and 2 (MFN1/2) were also assessed but similarly revealed no apparent interaction between them and GJA1-20k [56]. GJA1-20k is able to mediate mitochondrial fission through its ability to modulate the actin cytoskeleton. recruits actin around mitochondria independently of DRP1 and induces focal constriction sites leading to rapid fission events [56].
- Expand on the potential physiological significance of GJA1-20k dimerization. Speculate on how dimer formation might influence its interactions and functions within cells.
We thank the reviewer for their important suggestion. We have expanded our discussion on the physiological significance of GJA1-20k dimerization and how dimer formation could influence its interactions and functions within cells.
The revised manuscript has been updated to include the following (line 296 – 309):
Instead of a singular GJA1-20k protein linking different binding partners together, dimerization allows for distinct populations of GJA1-20k proteins to act as an intermediary between proteins without the risk of stereotactic interference impeding this interaction. This could explain how GJA1-20k is able to simultaneously rearrange the actin cytoskeleton, pattern microtubules in parallel arrays to actin, and ensure Cx43 vesicles are trans-ported along those microtubule highways to their corresponding membrane subdomains (Figure 2). GJA1-20k dimerization further facilitates actin recruitment around mitochondria and formation of constriction sites by having different GJA1-20k populations responsible for distinct roles.
The structure-function relationship of GJA1-20k is a prime example of how structural features of a protein can dictate its interactions and functional outcomes. The ability of GJA1-20k to form these dimers would allow it to participate in a wide range of cellular processes, reflecting its adaptability and importance in maintaining cellular function un-der conditions of stress.
- Suggest areas for future research, such as unexplored functions of GJA1-20k, its role in other cell types or tissues, and how understanding its mechanisms could lead to novel interventions for cardiovascular diseases.
We would like to thank the reviewer for this valuable comment. We have included in our manuscript a section detailing the clinical implications of GJA1-20k as a potential therapy in the setting of different diseases and organ systems. We highlighted studies that investigated the role of GJA1-20k in arrhythmogenic cardiomyopathy, traumatic brain injury, and cancer. We appreciate the reviewer’s concern to broaden the scope of our review to provide a more comprehensive understanding of the role of GJA1-20k in treatment and disease.
The revised manuscript has been updated to include the following (line 316 – 352):
- Clinical Considerations
Currently, GJA1-20k is primarily considered a trafficking protein essential for targeted delivery of Cx43. However, emerging evidence increasingly highlights its independent roles beyond its association with its parent protein. Understanding the mechanism of action of GJA1-20k is essential for the development of novel interventions for treating different diseases.
A recent study has explored the therapeutic potential for GJA1-20k in treating a desmoglein mutant murine model of arrhythmogenic cardiomyopathy (ACM) [74]. This study showed that GJA1-20k gene therapy can reduce arrhythmic burden in this ACM mouse model by rescuing impaired Cx43 forward trafficking, independent of systolic function. GJA1-20k restores Cx43 functionality, effectively reducing the risk of lethal arrhythmias associated with ACM. These results underscore GJA1-20k’s potential as a therapeutic target in cardiac arrhythmias and related pathologies.
Beyond the heart, emerging research suggests that GJA1-20k may play critical roles in other tissues and disease contexts. Studies have documented its role in rescuing mitochondrial networks under oxidative stress and providing neuroprotection after traumatic brain injuries (TBI). GJA1-20k protects mitochondrial networks under oxidative stress by acting as an organelle chaperone, leading to mitochondrial localization at the cell periphery [63]. This interaction was shown in astrocytes post-traumatic brain injury (TBI) wherein GJA1-20k stabilized mitochondria, facilitated their movement, and enhanced their transfer to neurons via tunneling nanotubules [75]. These findings suggest that GJA1-20k could be a therapeutic target for neurodegenerative diseases or brain injuries.
GJA1-20k has also been shown to regulate myometrial activity during labor. GJA1-20k expression is differentially regulated by progesterone receptor isoforms PR-A and PR-B which respectively promote and inhibit its activity. Dysregulation of GJA1-20k by hormonal perturbations may lead to preterm labor and other obstetric complications due to its downstream effects on Cx43 trafficking, gap junction formation, and myocyte connectivity, processes essential for synchronized uterine contractions [76].
The role of GJA1-20k in cancer pathology is also gaining attention. While full-length Cx43 has been widely studied as a tumor suppressor and, paradoxically, as a pro-metastatic agent in certain contexts, GJA1-20k may hold distinct regulatory functions [77,78]. Tishchenko et al. investigated the role of GJA1-20k in breast cancer cell behavior [79]. GJA1-20k was found to regulate actin organization, guide Cx43 delivery, and promote nanotubule formation, impacting proliferation, migration, and epithelial-to-mesenchymal transformation [79]. These findings indicate that GJA1-20k has a dual role in tumor sup-pression and progression, opening new research avenues for targeted cancer therapies.
- While the English is generally clear, a thorough proofreading could address minor grammatical errors or awkward phrasings to further improve readability.
We thank the reviewer for their feedback. We have simplified the language used and corrected grammatical errors to improve readability and clarity.

Reviewer 5 Report
Comments and Suggestions for Authors
In this paper, the authors provides a review of the internally translated isoform GJA1-20k of Connexin 43 (Cx43) and its potential roles in cardiac function and disease. The study explored GJA1-20k's involvement in Cx43 trafficking, cytoskeletal remodeling, and mitochondrial homeostasis, highlighting its significance in cardiovascular physiology and pathology. The following lists some comments. Firstly, it relies only on existing theoretical analysis, lacking the original experimental data introduction. The studies in the validation of GJA1-20k's roles in Cx43 trafficking and mitochondrial dynamics need be presented accordingly. Secondly, the authors need provide more information about the specific molecular interactions involving GJA1-20k and its effects on the cellular machinery. Additionally, the review could expand its scope by examining GJA1-20k's functions in tissues beyond the heart to provide a more comprehensive understanding of its role in disease pathogenesis and treatment.
Comments on the Quality of English LanguageEnglish need be polished.
Author Response
We thank the reviewer for the thorough review and valuable suggestions to improve our work. We have revised the manuscript and provided answers to the reviewer’s concerns to enhance the clarity and understanding of our work. We trust that our revised manuscript has adequately addressed the reviewer’s critiques. Please find below our detailed response to each specific comment.
In this paper, the authors provide a review of the internally translated isoform GJA1-20k of Connexin 43 (Cx43) and its potential roles in cardiac function and disease. The study explored GJA1-20k's involvement in Cx43 trafficking, cytoskeletal remodeling, and mitochondrial homeostasis, highlighting its significance in cardiovascular physiology and pathology. The following lists some comments.
- Firstly, it relies only on existing theoretical analysis, lacking the original experimental data introduction. The studies in the validation of GJA1-20k's roles in Cx43 trafficking and mitochondrial dynamics need be presented accordingly.
We would like to thank the reviewer for their insightful comments. We appreciate the reviewer’s suggestion to present original experimental data on the validation of GJA1-20k’s roles in Cx43 trafficking and mitochondrial dynamics. However, due to length constraints, we are unable to include the experimental details of the studies we referenced. We have instead referenced the original experimental studies that substantiate these findings. By referencing existing experimental studies throughout this review, our aim was to provide a comprehensive overview of the topic while ensuring our discussion is grounded in the original evidence. We hope this clarification addresses your concern, and we thank you once again for your valuable feedback.
- Secondly, the authors need provide more information about the specific molecular interactions involving GJA1-20k and its effects on the cellular machinery.
We thank the reviewer for this important feedback. While we appreciate the suggestion to include more detailed descriptions of the molecular mechanisms, we are unable to expand significantly on this aspect, as the current work on GJA1-20k’s interaction with the cytoskeleton to preserve mitochondrial function is an active area of work in our lab. We are optimistic that our current work will provide a more expansive response to these questions in the future. We aim to maintain a balance between providing a comprehensive overview and respecting the publication process for ongoing research.
- Additionally, the review could expand its scope by examining GJA1-20k's functions in tissues beyond the heart to provide a more comprehensive understanding of its role in disease pathogenesis and treatment.
We would like to thank the reviewer for this valuable comment. We have included in our manuscript a section detailing the clinical implications of GJA1-20k as a potential therapy in the setting of different diseases and organ systems. We highlighted studies that investigated the role of GJA1-20k in arrhythmogenic cardiomyopathy, traumatic brain injury, and cancer. We appreciate the reviewer’s concern to broaden the scope of our review to provide a more comprehensive understanding of the role of GJA1-20k in treatment and disease.
The revised manuscript has been updated to include the following (line 316 – 352):
- Clinical Considerations
Currently, GJA1-20k is primarily considered a trafficking protein essential for targeted delivery of Cx43. However, emerging evidence increasingly highlights its independent roles beyond its association with its parent protein. Understanding the mechanism of action of GJA1-20k is essential for the development of novel interventions for treating different diseases.
A recent study has explored the therapeutic potential for GJA1-20k in treating a desmoglein mutant murine model of arrhythmogenic cardiomyopathy (ACM) [74]. This study showed that GJA1-20k gene therapy can reduce arrhythmic burden in this ACM mouse model by rescuing impaired Cx43 forward trafficking, independent of systolic function. GJA1-20k restores Cx43 functionality, effectively reducing the risk of lethal arrhythmias associated with ACM. These results underscore GJA1-20k’s potential as a therapeutic target in cardiac arrhythmias and related pathologies.
Beyond the heart, emerging research suggests that GJA1-20k may play critical roles in other tissues and disease contexts. Studies have documented its role in rescuing mitochondrial networks under oxidative stress and providing neuroprotection after traumatic brain injuries (TBI). GJA1-20k protects mitochondrial networks under oxidative stress by acting as an organelle chaperone, leading to mitochondrial localization at the cell periphery [63]. This interaction was shown in astrocytes post-traumatic brain injury (TBI) wherein GJA1-20k stabilized mitochondria, facilitated their movement, and enhanced their transfer to neurons via tunneling nanotubules [75]. These findings suggest that GJA1-20k could be a therapeutic target for neurodegenerative diseases or brain injuries.
GJA1-20k has also been shown to regulate myometrial activity during labor. GJA1-20k expression is differentially regulated by progesterone receptor isoforms PR-A and PR-B which respectively promote and inhibit its activity. Dysregulation of GJA1-20k by hormonal perturbations may lead to preterm labor and other obstetric complications due to its downstream effects on Cx43 trafficking, gap junction formation, and myocyte connectivity, processes essential for synchronized uterine contractions [76].
The role of GJA1-20k in cancer pathology is also gaining attention. While full-length Cx43 has been widely studied as a tumor suppressor and, paradoxically, as a pro-metastatic agent in certain contexts, GJA1-20k may hold distinct regulatory functions [77,78]. Tishchenko et al. investigated the role of GJA1-20k in breast cancer cell behavior [79]. GJA1-20k was found to regulate actin organization, guide Cx43 delivery, and promote nanotubule formation, impacting proliferation, migration, and epithelial-to-mesenchymal transformation [79]. These findings indicate that GJA1-20k has a dual role in tumor sup-pression and progression, opening new research avenues for targeted cancer therapies.
